# Negotiation-Free Encryption for Securing Vehicular Unicasting Communication

**Fang Mei [1] , Shengjie Liu [1] , Jian Wang [1],* , Yuming Ge [2] and Tie Feng [1]**

[1] College of Computer Science and Technology, and Key Laboratory of Symbolic Computation and Knowledge Engineering of Ministry of Education, Jilin University, Changchun 130012, China; meifang@jlu.edu.cn (F.M.); liusj15@mails.jlu.edu.cn (S.L.); fengtie@jlu.edu.cn (T.F.)

[2] Technology and Standards Research Institute, China Academy of Information and Communications Technology, Beijing 100191, China; geyuming@caict.ac.cn

\* Correspondence: wangjian591@jlu.edu.cn; Tel.: +86-431-8516-8355

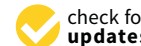

**Featured Application: Since the link changes rapidly in Vehicular Ad hoc Networks (VANETs), the security verification of the unicast communication between unfamiliar vehicles needs to shorten or even cancel the session establishment process for key negotiation, and reduce the bandwidth consumption. The method proposed in this work can improve the safety performance and transmission efficiency of VANETs, and will play an important role in the future intelligent transportation system. The existing cloud security mechanism can ensure the feasibility of this method.**

**Abstract:** Thanks to the rapid development of vehicle-to-everything (V2X) and sensor technology, states of vehicles can be accurately measured and stored jointly in the cloud. These states can be viewed as a set of infinite attributes, such as the density around the motor vehicle, signal strength and so on. As such, the vehicle can be viewed as a moving object. The vehicle state can be measured, and its entropy is large. In vehicle networking, unicast communications between vehicles must be encrypted. The previous approach was to negotiate a session key through the Diffie-Hellman algorithm and then use the session key to encrypt the communications. This method not only needs to know in advance the peer IP of the communication, but it also needs a long time to establish a session key. Therefore, it is not suitable for vehicle networking. For the fast-changing vehicle networking, the session key establishment process should be shortened or even canceled. In this paper, we propose a method of Negotiation-free encryption for securing vehicular unicasting communications to improve the efficiency of vehicle ad hoc network transmissions.

**Keywords:** vehicular ad-hoc network; unicast communication; encrypted transmission; property set; vehicle safety

---

## 1. Introduction

Vehicular Ad hoc Networks (VANETs) are special wireless networks which are designed to meet the needs of vehicular communications. Since applications of different regions have different characteristics, VANETs should be designed according to the characteristics of the vehicle to let the ad hoc network achieve the best performance in vehicle communications. Vehicles have a strong mobility characteristic compared with traditional self-organizing networks, usual encryption and authentication methods are not suitable for VANETs.

Because VANETs support safety services [1–5] which plays an important role in Intelligent Transport System (ITS), a high level of security is needed to ensure secure vehicular communications.

Therefore, data transmissions between vehicles are faced with some security issues. For instance, these include how to ensure that the data are not lost, stolen, or tampered with [6–8]. Additionally, there are some security concerns about privacy [9–11]. According to the present stage, end-to-end network transmission uses the Internet Key Exchange (IKE) Protocol. Before the transmission process, two stages are required: security protocols and key negotiations. After the Security Association (SA) negotiation, the process uses the DH algorithm for the key exchange transmission and distribution and then uses the AH and ESP protocols to encapsulate the package to ensure data security.

The IPSec protocol must negotiate all safety parameters, which contain encryption and identification algorithms [12], encryption and authentication keys, the key lifetime, etc. It costs a lot of overhead to ensure the establishment of a Security Association (SA). In addition, in the communication process, every time the packets are sent, the key negotiation will be reestablished. Thus, the process spends a significant amount of time negotiating and does not conform to the existing vehicle network environment. To achieve efficiency and safety for the vehicle's transmission network, we need to shorten the time spent on the negotiation before the data transmission and reduce the average delay and the bandwidth overhead of the data transmission as much as possible without compromising security. Thus, we propose a negotiation-free encryption method (NFEM) to reduce the preparatory work before the data transmission and then achieve high efficiency and safety. In today's vehicular environment, a trustworthy cloud and OBU (on-board unit) are easily to implement, which provides our work with a reliable premise.

As for the NFEM, we use the vehicle state attribute value as a packet encryption key to replace the traditional IPSec key negotiation and management stage. How to address the vehicle properties is a key point in this paper. We use a trustworthy cloud to manage the real-time status of every vehicle. Before transmitting, the sender requests the cloud for the receiver's state value, which is used to encrypt the transmitted data. Compared with IPSec, this method reduces the overhead of the transmission process, such as policy consultations, DH value consultation, nonce exchange, authentication, the cryptographic algorithm, some of the necessary auxiliary data, etc. Negotiation and authentication are particularly complex processes. However, in the NFEM, the vehicle simply needs to request the historical state value of itself from the cloud, which is equivalent to obtaining the decryption key and does not require the parties to negotiate. Thus, it considerably reduces the negotiation and validation overhead. To prevent the vehicle state privacy from leaking, the cloud will hash the status value before the data are transmitted. Such measures will prevent attackers from intercepting the vehicle's state value and prevent malicious requests that aim to know other vehicles' status value. In the process of messaging, we use the timestamps as the time recording of the messaging. Vehicles only can request their own historical status values and the real-time status values of other vehicles, which naturally prevents MITM (Man-in-the-Middle) attacks. Even if the attacker intercepts the transmission data, the ciphertext cannot be successfully decrypted, since the timestamps cannot be tampered with. If you do not change the timestamp, there is no permission to obtain information about the state of the history of others. Otherwise, you cannot get the correct time state value, and thus the decryption fails.

We mainly discuss how the data of end-to-end vehicle transmissions in the VANET environment can be transmitted more efficiently and safely, and then we propose a more suitable encryption method for VANET. We use the state value of the vehicle to encrypt the plaintext to reduce transmission delay. With respect to the bandwidth overhead, compared with the traditional large number of loads, requirement of the bandwidth is less in our method. In terms of security, we use the cloud mechanism and the unchangeable timestamps to ensure the data security and privacy.

The rest of this paper is organized as follows. Section 2 provides a summary of the relevant work, and Section 3 introduces the system model and thread model used in this paper. Section 4 describes the details of the NFEM. Section 5 analyzes the feasibility and safety of the NFEM. Section 6 shows the encryption method performance analysis and simulation. Section 7 draws a conclusion and offers ideas for future work.

## 2. Related Work

VANET plays an important role in futrue Intelligent Transport System ITS, secure trade-off between vehicles is very important as VANET supports safety-related applications. Because topology of the VANET changes rapidly, it is challenging to ensure real-time and security data transmission. A lot of techniques have been proposed to ensure secure privacy-preserving vehicular communications.

Sun et al. [13] proposed an efficient pseudonymous authentication scheme with strong privacy preservation (PASS) for VANETs. A novel scheme based on one-way hash chain is designed to generate the pseudoidentities of the pseudonymous certificates belonging to the same owner, and an efficient certificate-updating scheme is also proposed. Hao et al. [14] proposed a distributed key management framework based on group signature to provision privacy in vehicular ad hoc networks (VANETs). An efficient cooperative message authentication protocol is developed to reduce the computation and communication overhead.

Zhang et al. [15] proposed a decentralized authentication protocol which use the RSUs to establish a communication group and nodes belong to the group request secret member keys from the RSUs. Huang et al. [16] proposed pseudonymous authentication-based conditional privacy (PACP) scheme, RSUs are also used to generate pseudonyms for anonymous communication. PACP allows vehicles to generate provably anonymous and computationally efficient pseudonyms to ensure conditional privacy. Pandi et al. [17] proposed a secure dual authentication technique with the capability of preventing malicious vehicles entering into the VANET system. A dual group key management scheme is also proposed to efficiently distribute a group key to a group of users and to update such group keys during the users' join and leave operations.

Wasef et al. [18] proposed an Expedite Message Authentication Protocol (EMAP), which replaces the time consuming CRL checking process by an efficient revocation checking process. EMAP also uses a novel probabilistic key distribution, which enables non-revoked OBUs to securely share and update a secret key.

Lin et al. [19] proposed a cooperative authentication scheme that does not involve inter-vehicle interaction, an evidence-token mechanism is added to resist the free-riding attacks that do not use fake authentication efforts.

Wang et al. [20] proposed a lightweight and efficient strong privacy preserving (LESPP) authentication scheme by mainly using message authentication code (MAC) and symmetric encryption. The proposed scheme can reduce both computation and communication overhead. Sign messages which were used to sign in identity-based signatures can be omitted, this can further reduces communication overhead and avoids certificate management.

He et al. [21] proposed a new ID-based CPPA scheme for VANETs without using bilinear pairing, which provides the function of batch verification of multiple messages. The proposed scheme does not use bilinear paring but still supports both the mutual authentication and the privacy protection simultaneously. Li et al. [22] proposed a novel framework with preservation and repudiation (ACPN) for VANETs, in which a public-key cryptography (PKC) to the pseudonym generation is introduced, which ensures legitimate third parties to achieve the non-repudiation of vehicles by obtaining vehicles' real IDs. The existing ID-based signature (IBS) scheme and the ID-based online/offline signature (IBOOS) scheme are used, for the authentication between the road side units (RSUs) and vehicles, and the authentication among vehicles.

These schemes lowered time costs in computational delay of the V2X authentication, and higher efficiency can be achieved. However, there are still many drawbacks to be improved (the complexity of the key negotiation, the costs of security certificate management, the long delays in negotiations, etc.), and the unique features of Vehicle Networking have not been effectively used. Our proposed NFEM uses vehicle state attribute value as packets encryption keys. The security cloud is used to store the vehicular state values. As a new generation of infrastructure-based wireless technologies, we use Long Term Evolution (LTE) to transfer request packets and response keys packets between mobile vehicles

and clouds in our NFEM architecture. There are already some standars and application, indicating that LTE is suitable for vehicular communicaition or vehicle-to-cloud communication.

Thanks to its simplified flat all-IP architecture, LTE can provide a round-trip time theoretically lower than 10 ms, and transfer latency in the radio access up to 100 ms. LTE is very beneficial for delay-sensitive vehicular applications, and there is already a standard that is considered as the preliminary version of 4G mobile communications [23]. The scenarios are suitable for operating LTE-based V2X services and address the main challenges of high mobility and densely populated vehicle environments in designing technical solutions to fulfill the requirements of V2X services. By leveraging the spectral-efficient air interface, the cost-effective network deployment, and the versatile nature of supporting different communication types, LTE systems along with proper enhancements can be the key enabler of V2X services [24]. The long-term evolution-vehicle (LTE-V) standard for sidelink or V2V communications based on the PC5 interface includes two radio interfaces. The cellular interface (named Uu) supports vehicle-to- infrastructure communications, while the PC5 interface supports V2V communications based on direct LTE sidelink [25]. The European AutoMat project defines an open Common Vehicle Information Model (CVIM) in combination with a cross-industry, cloud-based big data marketplace. A simulative analysis of car-to-cloud data traffic is presented by setting up a car-to-cloud communication model, leveraging LTE uplink channels, which is founded on a measurement-based empirical channel model. CVIM Data Packages are sent to the cloud leveraging LTE sidelink. The results quantify the available data rate for car- to-cloud communication and a vehicle traffic state aware data aggregation is proposed [26].

The main contributions of the literature can be summarized as follows.

1. A Negotiation-free encryption way is proposed, which is customized for VANETs.
2. The vehicle state attribute value is used as a packet encryption key to replace the traditional IPSec key negotiation and management stage which is unique and difficult to forge.
3. The secure cloud mechanism and the unchangeable timestamps are used to ensure the data security and privacy.

The proposed scheme can considerably reduce the negotiation and validation overhead. The scheme performs well even under the condition of a high density of vehicles and a high frequency of beacon conditions.

## 3. Modeling

In the unicast transmission process, IKE is used to exchange and manage the encryption key, which solves the problem of safely establishing or updating the shared key in an insecure network environment such as the Internet. With the development of VANET, we found that the key negotiation takes up a lot of time in the transmission process. Compared to the traditional unicast transmission method, the NFEM improves this problem. This section mainly builds the overall system model of the NFEM and enumerates the possible attack model.

### 3.1. System Model

From Figure 1, we can see that the model in this paper consists of three entities: Certification Authority (CA), Vehicular Cloud (VC), and On-board Unit (OBU).

1. CA: The CA provides a certificate for each user who has a public key. It is used to identify the true identity of the electronic certificate holder. Additionally, it will prove that the user has a legitimate identity to use the public key. In the data transmission process, CA ensures the authenticity of the vehicle and the cloud, and it also prevents illegal vehicles changing data in this process.
2. VC: The VC is a vehicle cloud agency. Real-time information of each legal vehicle is stored in the Vehicular Cloud, and it will provide a timely response when vehicles make a request for status information. In addition, the VC must hash status information in order to avoid the loss of

privacy when a status inquiry is requested. The obtained hash value represents the state of the vehicle at time t.

3.  OBU: Each vehicle has an on-board unit (OBU). The OBU is comprised of a hardware security module that is responsible for encrypting the data, storing the status information, positioning and other operations.

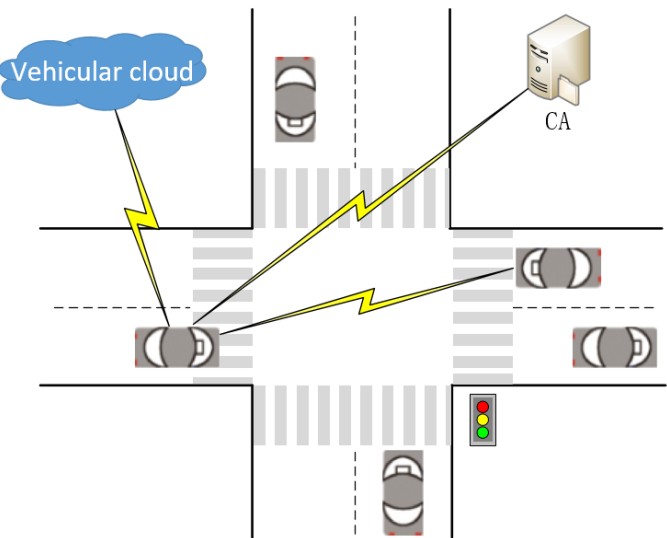

**Figure 1.** The system model.

*3.2. Threat Model*

With the development of the existing vehicle ad hoc network, the wireless network transmission process with regard to security also encounters increasingly more security problems and challenges. The notable characteristic of VANET is related to the driver's life. With these threats and attacks in the VANET, how to avoid data eavesdropping, tampering and other operations becomes a great challenge. The transmission method will face attacks, which are summarized in the following types.

1.  Message stealing: In the message transmission process, an attacker can steal others' messages by monitoring. Once the message has high confidentiality, it will cause great losses for senders and receivers.
2.  Privacy attacks: According to the stolen sensitive information, attackers can analyze drivers, such as driving habits, and this can cause security risks for the drivers.
3.  Fake news attack: This passes fake messages to other vehicles in order to achieve illegal purposes.
4.  Fake identity attack: This forges fake identities to conceal their real ones to pass messages.
5.  Non-repudiation attack: In the vehicle transmission process, if someone maliciously sends some false information to others when tracing the sources for the sender, it has non-repudiation.
6.  Replay Attack: A message is resent to the vehicle cloud in order to achieve the role of deception.
7.  Dos attack: It sends a large amount of data to plug the communication channel in order to cause a channel jam or prevent vehicles from accessing the network service.

**4. Negotiation-Free Encryption Method**

In this section, we describe how the NFEM can be implemented. The whole process includes three stages: key generation, data encryption transmission and decryption authentication. The key negotiation is no longer needed, and we use the permission, which is used to acquire the state of the vehicle to control the encryption and decryption. In the communication process, the communicated vehicles interact with vehicle cloud by LTE channel only once respectively, saving a lot of repeated

interaction steps. LTE is very beneficial for delay-sensitive vehicular applications, which can provide a round-trip time theoretically lower than 10 ms, and transfer latency in the radio access up to 100 ms.

### 4.1. Prepare Conditions

Table 1 defines the main parameter symbols used in this article.

**Table 1.** The main symbol definition.

| Definition | Description |
|---|---|
| $ID_s, ID_r$ | Send message Vehicle ID, Receive message vehicles ID |
| $V_s, V_r$ | Send message Vehicle, Receive message vehicles |
| $VC$ | Vehicular Cloud |
| $TS$ | Timestamp |
| $(V_{s_{pk}}, V_{s_{sk}}), (V_{r_{pk}}, V_{r_{sk}})$ | Sender's Public and private keys, Receiver's Public and private keys |
| $VC_{pk}, VC_{sk}$ | VC's Public and private keys |
| $Attr_{r_i}$ | Receiver's attribute i |
| $Hash(Attr)$ | The hash value of state property |
| $En_{key}$ | Use the key to encryption |
| $Data$ | Plaintext |
| $key$ | Sender used to encrypt data |

### 4.2. Algorithm Implementation

#### 4.2.1. Key Generation

In VANET, the vehicle communications are confidential. Before passing the message, we must generate the encryption key to guarantee the security of the message. Therefore, the receiver is able to decrypt the encrypted data efficiently. Here, we use the receiver's state value ($Attr_{r_i}$) as the encryption key. To prevent a vehicle privacy leak, the vehicle cloud (VC) will hash the value before transmitting the state value and then sends the hash state value ($Hash_{Attr}$) back to the requesting vehicle. The key generation process is shown in Figure 2.

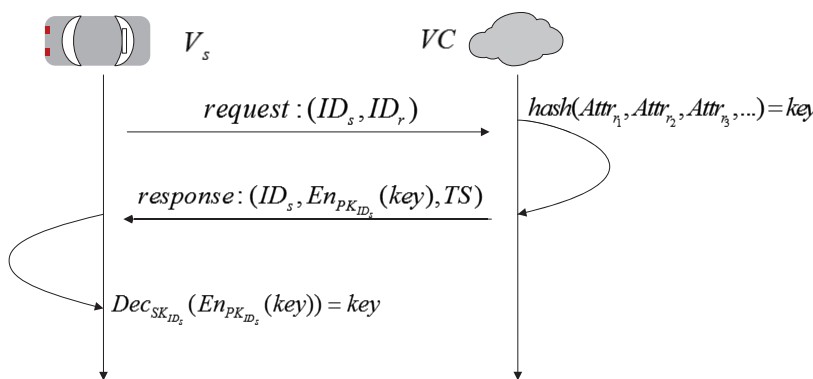

**Figure 2.** Key Generation.

1.  $V_s$ sends the request of Receiver's $V_r$ status value to the $VC$. We use public key of $VC$ to encrypt $Message_{req}$. The format of the request message is as follows:

$$Ciphertext_{(V_s-to-VC)} = Enc_{VC_{pk}}(ID_s + ID_r) \tag{1}$$

2. The $VC$ first decrypts the $Ciphertext_{(V_s-to-VC)}$ with its own Secret Key.

$$Message_{rec} = Dec_{VC_{sk}}(Ciphertext_{(V_s-to-VC)}) \qquad (2)$$

Then it find $V_r$'s real-time state $(Attr_{r_1}, Attr_{r_2}, Attr_{r_3}, ...)$ and hashes them. This hash value is the key which used to encrypt message when $V_s$ transmits the data to $V_r$.

$$key = Hash(Attr_{r_1}, Attr_{r_2}, Attr_{r_3}, ...) \qquad (3)$$

3. The Cloud uses $V_s$'s public key to encrypt the *key*. The encrypted data and the real time as time-stamp ($TS$) will be sent to $V_s$. The format of the return message is as follows:

$$Ciphertext_{(VC-to-V_s)} = Enc_{V_{s_{pk}}}(ID_s + key + TS) \qquad (4)$$

4. $V_s$ will use its own private key to decrypt the encrypted data and get the real key for encryption.

$$key + TS = Dec_{V_{s_{sk}}}(Ciphertext_{(VC-to-V_s)}) \qquad (5)$$

4.2.2. Data Encryption Transmission

After getting the key, the system uses the key and timestamp to encrypt the data to ensure the transmission security. The format of the message is shown in Figure 3.

$$Message = ID_r + Enc_{key}(ID_s + data) + TS \qquad (6)$$

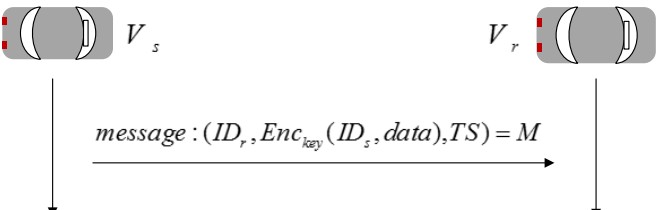

**Figure 3.** Data transmission.

$V_s$ uses a key to encrypt the *Data* and gets the ciphertext. Then $V_s$ transmits the ciphertext and the time-stamp to $V_r$.

4.2.3. Decryption Authentication

$V_r$ receives the ciphertext and the timestamp. Now it is time to decrypt the ciphertext. However, the current time is later than the timestamp. Therefore, the decryption key is its own history of the state. For the state value of the history, only the vehicle itself can submit a request to the cloud, but others cannot. The decryption authentication steps are shown in Figure 4.

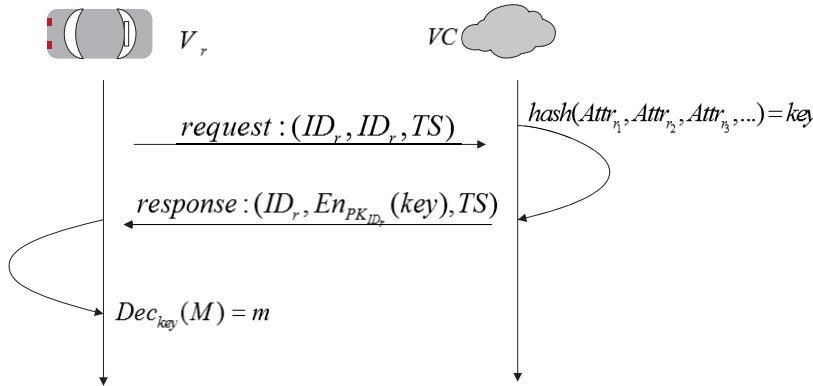

**Figure 4.** Decryption authentication.

1.　$V_r$ sends the request for its status value to the $VC$ and the time of the request state is the same time of timestamp time ($TS$).

$$Ciphertext_{(V_r-to-VC)} = Enc_{VC_{pk}}(ID_r + ID_r + TS) \tag{7}$$

2.　The $VC$ first decrypts the $Ciphertext_{(V_r-to-VC)}$ with its own Secret Key.

$$Message_{rec} = Dec_{VC_{sk}}(Ciphertext_{(V_r-to-VC)}) \tag{8}$$

Then it finds $V_r$'s historical states $(Attr_{r_1}, Attr_{r_2}, Attr_{r_3}, ...)$ and hashes them. The hash value has a *key* that is used to decrypt the received ciphertext.

$$key = Hash(Attr_{r_1}, Attr_{r_2}, Attr_{r_3}, ...) \tag{9}$$

3.　The cloud uses $V_r$'s public key to encrypt the *key* and the timestamp ($TS$) and then sends it back to $V_r$.

$$Ciphertext_{(VC-to-V_r)} = Enc_{V_{r_{pk}}}(ID_r + key + TS) \tag{10}$$

4.　$V_r$ uses its private key to decrypt the ciphertext to get the real *key*.

$$key + TS = Dec_{V_{r_{sk}}}(Ciphertext_{(VC-to-V_r)}) \tag{11}$$

5.　$V_r$ uses the key to decrypt the ciphertext ($M$) and receives the message.

$$Data = Dec_{key}(Message) \tag{12}$$

## 5. Feasibility and Safety Analysis

We have introduced the general model of the NFEM. In this section, we analyze the safety and feasibility of our proposed method, and the results show that the NFEM is suitable for vehicular networking applications.

### 5.1. Feasibility of No Key Negotiation

Message transmission security can be ensured even without the key agreement process by using the NFEM. Messages can be decrypted by recipients by using the state value of the vehicle as the key. The state value is the most distinctive property of the moving vehicle, as vehicles have different state values at different times and in different positions. These state values are divided into two types: the dominant state value and the implicit state. The dominant state values include the vehicle color,

brand and other known properties. The implicit state values cannot be known by others, including the vehicle's own internal attributes and status. This can ensure that the state value can be used as a key. If one vehicle wants to encrypt a message using the other vehicle's state value, they must first obtain the hashed state values of that car from the trusted cloud. One vehicle can get its own historical state value from the trusted cloud and choose one as the decryption key according to the timestamp of the received message. Therefore, before communications, two vehicles can also use a key to encrypt and decrypt without the key negotiation.

*5.2. Security Analysis*

**Lemma 1.** *The Cloud stores a large amount of vehicles' state information, and it is characterized by accuracy, timeliness, confidentiality, integrity, etc.*

**Proof of Lemma 1.** The Negotiation-free encryption algorithm depends on the state information of the vehicle. Therefore, there is a high requirement for the accuracy, timeliness, confidentiality and integrity of the state information. Thanks to the rapid development of V2X and sensor technology, the state of the vehicle can be accurately measured and stored in the cloud, which ensures the accuracy of the data. When the vehicle-end requests the data in the cloud, it will follow a mechanism through which vehicles can check their own history state and can also request the real-time state of other vehicles. However, one cannot query the historical state of other vehicles. Thus, one can use different permissions of the vehicle state to ensure the confidentiality of the information transmission. In the Negotiation-free encryption algorithm, the establishment process of the session key can be shortened or even canceled. To a large extent, this ensures the timeliness of message transmission. These states can be viewed as collections of infinite attributes, such as the density of the vehicle around vehicles, the signal strength, etc. Therefore, the vehicle can be viewed as a mobile object in which its state can be measured, and the entropy of the state value is very large. The integrity of information transmission can be ensured by using signatures to identify the source of the public key to prevent attacks from middlemen.  □

**Lemma 2.** *For the passive attacker, there is no security problem.*

**Proof of Lemma 2.** No matter whether the cloud uses its own private key or a shared public key to decrypt the message sent by the vehicle that uses the public key to encrypt the query request, it fully complies with the existing public key encryption protocol. For passive attackers, there will be no security issues at all.  □

**Lemma 3.** *For the active Man-in-the-Middle Attack, it is very difficult to decrypt the ciphertext in vehicle-to-vehicle communications when an attacker intercepts the message.*

**Proof of Lemma 3.** This encryption algorithm proposed by us naturally prevents the active Man-in-the-Middle Attack. When vehicle A send a message to vehicle B, the timestamp and vehicle B's real-time state value are used to encrypt the message. If an attacker intercepts the message and modifies the timestamp, the ciphertext information will not be decrypted, since the correct state value cannot be obtained.  □

**Lemma 4.** *Even if the attacker intercepts the ciphertext and obtains the timestamp, he is still unable to decrypt the ciphertext.*

**Proof of Lemma 4.** The ciphertext between vehicles is encrypted based on the historical state of the receiver. At this time, for any users, only the receiver can query the value of its own history state from the cloud. This means that others cannot obtain the prerequisites to decrypt the ciphertext, and thus the problem is solved.  □

**Lemma 5.** *The attacker uses the exhaustive method to attack the ciphertext through brute force, but the algorithm is still in a safe state.*

**Proof of Lemma 5.** If attackers use brute force to decrypt the ciphertext, they have to traverse all the possible status values until they find the right one. However, the Negotiation-free encryption has a great feature in that the entropy value of the key space is very large. These states can be considered as a collection of infinite properties around the vehicle, such as the vehicle density, signal strength, etc. The state can be measured while the entropy value of the state is large. That is, the value range is large and cannot be completed at all for a limited time. □

## 6. Performance Evaluation

This section is divided into two parts. We analyze the Negotiation-free encryption algorithm and traditional IPSec first and then realize the simulation experiment using the network simulation tool GNS3.

We summarize the characteristics of several encryption algorithms of NFEM and IPSec, to indicate the superiority of NFEM. The comparisons are shown in Table 2.

**Table 2.** Algorithm comparison.

| Characteristic | NFEM | IPSec |
|---|---|---|
| Key distribution Center | Vehicle Cloud | KDC server |
| Channel setup | No need | IKE negotiation to establish SA |
| Key Generation | Hash Corresponding Vehicle Status | 3DES192, AES128 or AES256 |
| Key distribution | Both vehicles get key from cloud separately | Requestor handover the key to respondent |
| Decryption consumption | Symmetric decryption algorithm | depends on key length |

In the experiment, we compare the two respective methods using the security encryption delay and the transmission overhead. Then, we analyze the NFEM separately using the ns-3 (version 3.26) simulator. We analyze the performance of the Negotiation-free encryption algorithm according to the vehicle density, vehicle speed and beacon transmission frequency. The main parameters are shown in Table 3.

**Table 3.** Simulation Parameters.

| Simulation Parameters | Value |
|---|---|
| $Car\ ID_{req}$ | 5 bytes |
| $Car\ ID_{res}$ | 5 bytes |
| $State\ Value\ (hash)$ | 32 bytes |
| $Message\ size$ | 20 bytes |
| $Ciphertext\ size$ | 45 bytes |
| $Ciphertextsize$ | 2 bytes |

### 6.1. Comparison and Analysis of Two Methods

Our experimental environment is the following: 64-bit 3.6 GHz Core i7 processor with 4G memory. The experimental simulation environment includes a GNS3. We select the AES256 encryption method for the comparisons.

According to the transmission mechanism, this paper defines the average delay of the message as follows:

$$Delay = \frac{1}{N} \sum_{i=1}^{n} (T_{KeyGen} + T_{KeyAuth}) \tag{13}$$

N is the number of messages, $T_{KeyGen}$ is the time which is spent to generate the encryption key and $T_{KeyAuth}$ is the time that is spent to generate the decryption key.

$$T_{KeyGen} = T_{req} + T_{hash} + T_{res} \tag{14}$$

$T_{req}$ is the time that the sender requests the state of the receiver. $T_{hash}$ is the time that the cloud spends on hashing the state value, and $T_{res}$ is the time that the cloud spends on sending the hash value of the receiver to the sender.

$$T_{KeyAuth} = T_{req} + T_{res} \tag{15}$$

$T_{req}$ is the time that receiver requests its state value from the cloud. $T_{res}$ is the time that the cloud spends on sending the hash value of the receiver's state value to the receiver.

For the Negotiation-free encryption algorithm, the average delay of the transmission is 150 ms. The major consumption steps are the key initialization and the authentication process, which are respectively 61 ms and 62 ms.

In the Negotiation-free encryption algorithm process, there are some other costs. For example, the average cost of the hash of the vehicle state using SHA256 is 0.016 ms. It encrypts the plaintext using AES and the 256-Bit hash value as the encryption key. The average overhead is 3 ms. Because they occupy a small portion of the total time, which can be negligible, this paper will only take two parameters ($T_{KeyGen}$ and $T_{KeyAuth}$) as the main overhead, and the rest can be ignored.

After many IPSec simulation experiments, the average delay is 310 ms. The main time-consuming processes are the D-H method calculation, the certification of digital signatures and the consultation process. The delay of these three parts is 35 ms, 80 ms and 170 ms, respectively. In the IKE encryption process, the consumption of other parts is among 0 to 1.5 ms, such as the encryption, decryption, hashing generation and authentication, and the digital signature authentication. All these can be neglected since they occupy only a small portion of these costs. According to the NFEM, the average time of key generation is 61 ms and the average time of decryption and authentication is 62 ms. Added together with the encryption and transmission delay, the total delay is 150 ms. Compared with the average delay of 310 ms of IPSec, our method saves half of the time. Therefore, on the aspect of the average delay, the NFEM is better than IPSec.

According to the different encryption methods, Table 4 compares the average delay.

**Table 4.** Average delay.

| Transmission Method | 3DES192 | AES128 | AES256 |
|---|---|---|---|
| IPSec | 309.7 ms | 311 ms | 310 ms |
| Negotiation-free encryption method | 149.8 ms | 150.5 ms | 150 ms |

We can summarize that the efficiency of the Negotiation-free encryption transmission is twice that of the negotiation mechanism of the traditional IPSec.

We define the formula of average transmission bandwidth as

$$bandwidth = \frac{1}{N} \sum_{i=1}^{n} \left( \frac{DataLenth}{Delay} \right) \tag{16}$$

N is the number of messages, $DataLength$ is the data length in transmission, and Delay is the average delay.

Then, we compared the performance of our proposed Negotiation-free encryption method and IPSec, the result is shown in Table 5. We send a 20-byte payload with 411 bytes of data using IPSec. The average delay is 310 ms and the average bandwidth is 1.3 bytes/s. Using the NFEM, we also send a 20-byte payload with 127 bytes of data and the average delay is 150 ms. Therefore, the average bandwidth is 0.84 bytes/s. Obviously, the NFEMs greatly reduce the bandwidth overhead.

**Table 5.** Average bandwidth.

| Transmission Method | Bandwidth |
|---|---|
| IPSec | 1.3 byte/s |
| Negotiation-free encryption method | 0.84 byte/s |

### 6.2. Performance Evaluation of NFEM

Here, we evaluate the performance of our proposed NFEM. In this paper, the key negotiation process is canceled by using the vehicle attribute as a message encryption key stored in the cloud to replace the key negotiation process. Therefore, throughout the transmission, we can divide the whole process into two parts. One is the transmission process between the vehicle and the cloud, and the other is the transmission process between vehicles. Then, we focus on these two parts to analyze the performance of the whole transmission process.

### 6.2.1. Simulation Environment

The road network represents the urban scenario of a 5 × 5 grid with 25 blocks, as illustrated in Figure 5. There are six vertical and six horizontal two-lane roads with 200 m spacing between them. To study the impact of vehicle density, the numbers of vehicles are varied from 25 to 125 with an increment of 25. Similarly, to study the impact of speed, the average speed of the vehicle is varied from 20 to 100 km/h with an increment of 20 km/h. The transmission range is 200 m and vehicles are deployed over an area of 1 km × 1 km. The relative simulation parameter is shown in Table 6.

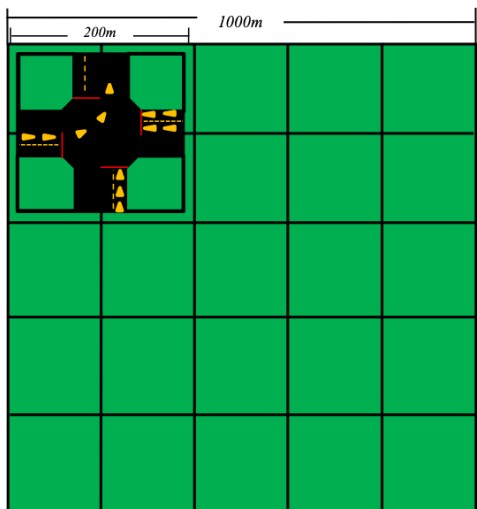

**Figure 5.** Road network, 5 × 5 Manhattan grid.

**Table 6.** Simulation parameters.

| Simulation Parameter | Value |
|---|---|
| Number of vehicles | 25, 50, 75, 100, 125 |
| Vehicle average speed | 20, 40, 60, 80, 100 km/h |
| Beacon transmission frequency | 1, 4, 8, 10, 20 Hz |
| Simulation area | 1000 m × 1000 m |
| Simulation duration | 50 s |
| packet size | 512 B |

With the continuous breakthroughs of VANETs, some excellent vehicular communication simulators are produced, such as TraNS, Veins and iTETRIS etc. These popular simulators work well in simulating vehicle behavior mode and emergency intervention. In our experimental scenario,

we divide the transmission process into two parts. We mainly measure the request and return delay of key packages in vehicle-to-cloud communication, and the delay of data transmission between vehicles. As a widely used open source simulation platform, NS-3 provides many integrated modules for different test requirements, including LTE-EPC Network Simulator (LENA) and Wireless Access in Vehicular Environment (WAVE). The architecture of LENA module includes LTE model and Evolved Packet Core(EPC) model, of which LTE model provides Ratio Access Network (RAN) protocol stack and EPC provides Core Network (CN) protocol stack. WAVE module supports IEEE1609 standard and can provide V2V and V2I communication. Considering the complexity of LTE system and the specificity of vehicle cloud communication simulation requirements, LENA can meet our LTE side test requirements. In the transmission process between vehicles, we use the WAVE module for ns-3. By default, each vehicle transmits 512 B beacons at varying transmission frequencies using an UDP-based application. The simulation parameters and values are shown below.

6.2.2. Impact of Varying Beacon Transmission Frequency and Speed

Figure 6 shows the performance of the Negotiation-free encryption algorithms in terms of delay with different beacon transmission frequencies and vehicle densities. In this figure, the average speed is 60 km/h. As we can see, when the vehicle density is less than 75 or the beacon transmission frequency is less than 8 Hz, the transmission delay is relatively small. With the increase of beacon transmission frequency and vehicle density, the delay increased.

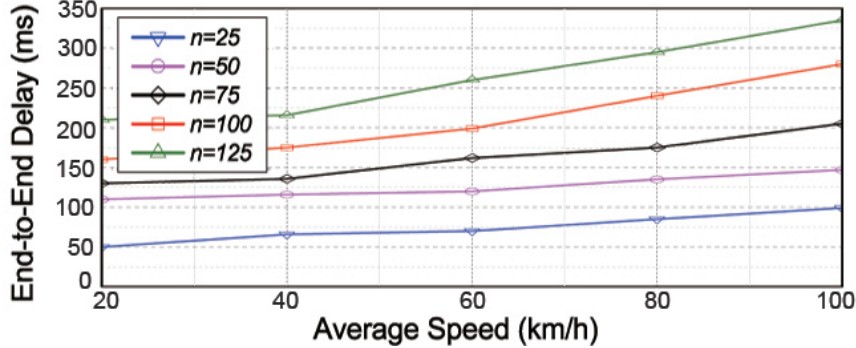

**Figure 6.** End-to-end delay (ms) vs. beacon transmission frequency (Hz).

Figure 7 shows the performance of the Negotiation-free encryption algorithms in terms of delay with different beacon transmission frequencies and vehicle densities. In this figure, the beacon transmission frequency is 8 Hz. Generally, with the increase of the average speed and vehicle density, the delay rises. Sparser topologies satisfy the delay requirements. When the vehicle density is less than 75, the delay is basically below 200 ms.

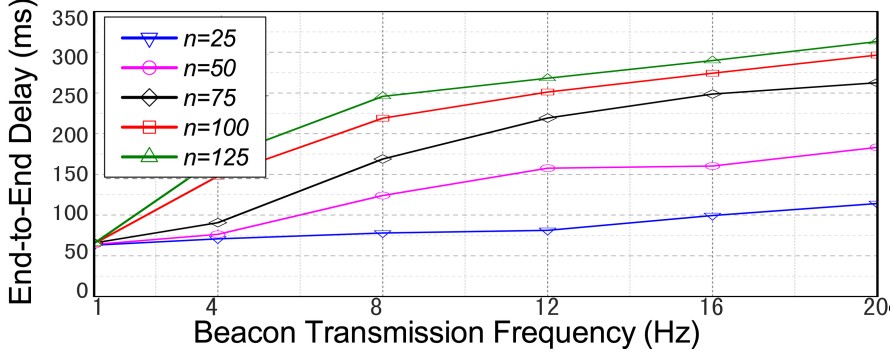

**Figure 7.** End-to-end delay (ms) vs. Average Speed (km/h).

Using the Negotiation-free encryption algorithm, the percentage of delays in the entire transmission process by the WAVE and LTE technology is calculated according to the different beacon transmission frequencies. To see the entire transmission process and the delay proportion of different technologies, we set the simulated numbers of vehicles between 25 and 100, respectively. In Figure 8, the vehicle density is 25, and in Figure 9, the vehicle density is 100.

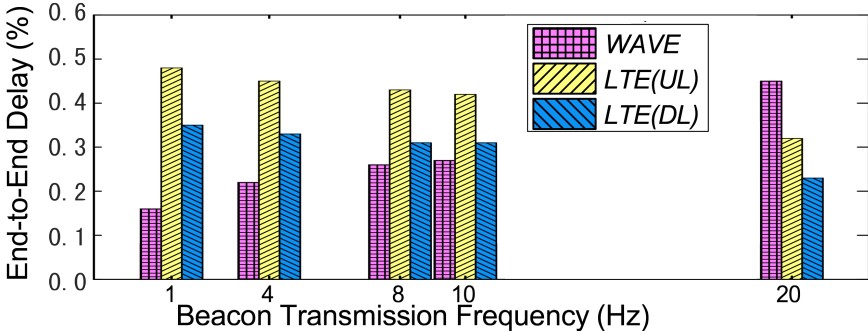

**Figure 8.** Beacon Transmission Frequency vs. delay (a).

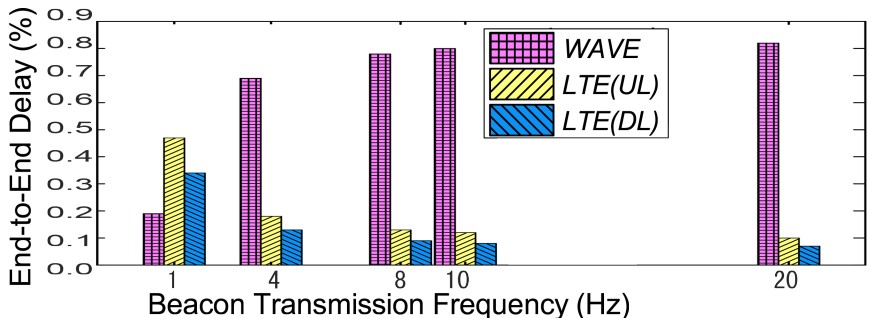

**Figure 9.** Beacon Transmission Frequency vs. delay (b).

From Figures 8 and 9, one can clearly see that the beacon transmission frequency has little effect on the delay of the LTE compared with WAVE in the transmission process of the Negotiation-free encryption algorithm. With the increase of the beacon transmission frequency, the delay between vehicles increases rapidly. Therefore, in the whole transmission process, the increase of the delay proportion of the Negotiation-free encryption algorithm is obvious.

Experiments show that our Negotiation-free encryption algorithm transmission method can adapt well to the actual needs and performs well under the condition of a high density of vehicles and a high frequency of beacon conditions.

## 7. Conclusions

This paper proposes the negotiation-free encryption method based on VANETs. In NFEM, the existing vehicle status information stored on the cloud can be used to generate the encryption key for each vehicle communication session, and the vehicle cloud is regarded as a key distribution center, with no need to build a new key distribution server. Moreover, the performance of cloud is also far superior to that of traditional centralized key distribution servers.

From the process point of view, the traditional KDC generates the encryption key for the communication session, and encrypt and send it to the requestor. Then the requestor transfers the key to the respondent. When the respondent receives the encryption key, it also needs to send a reply to the requestor to verify that the received communication request is true. During this period, there are several interaction steps between the two sides of communication, and there exists a risk of being attacked by a third party. In the NFEM method, the cloud server generates the encryption

key by hashing the vehicle status information of the respondent at a certain time, and returns the key to the both communicating parties separately. Based on the timestamp, the respondent obtains the encryption key directly from the vehicle cloud, decrypts the ciphertext, which avoids the extra interaction steps of KDC and the third party attack risk.

Compared with the traditional encryption methods of self-organizing network, NFEM mainly has the following characteristics. (1) In the transmission encryption process, it can reduce the key agreement, delay of encryption and overhead of key management. (2) The state value of vehicle attribute is used as a key, rather than by calculation and consultation with some of the key independent attributes of the vehicle itself, which greatly reduces the key generation time and meets the application requirements of VANETs. (3) In contrast with the traditional consultative mechanisms, NFEM reduces bandwidth consumption. In complex distributed vehicles network environments, as the vehicle density increases, the minimized bandwidth usage will be more favourable to network transmission. (4) In terms of security, the invariability of vehicle cloud and timestamp can ensure the security and privacy of data communication and control transmission for VANETs. Hence, our next work will be focused on transportation security.

**Author Contributions:** F.M., J.W., and Y.G. designed the study. F.M., S.L., and J.W. developed the computer simulations and performed the data analysis. F.M., T.F. prepared the figures and tables. F.M. and J.W. wrote the final manuscript. All authors read and approved the final manuscript.

**Funding:** This research was funded by the National Nature Science Foundation of China Grant Number 61373123, 61572229, U1564211, 6171101066 and 6187060366, by the Jilin Provincial Science and Technology Development Foundation Grant Number 20170204074GX and 20180201068GX, by the Jilin Provincial International Cooperation Foundation Grant Number 20180414015GH, by CERNET Innovation Project Grant Number NGII20180701.

**Acknowledgments:** The authors would like to thank the Associate Editor and the reviewers for their valuable comments that helped to improve the quality of this paper.

**Conflicts of Interest:** The authors declare no conflict of interest.

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
