# Peer review of "Negotiation-Free Encryption for Securing Vehicular Unicasting Communication"

_applsci, doi:10.3390/app9061121_

Round 1
Reviewer 1 Report
This paper proposes a protocol for securing unicasting communication called Quantum-Inspired Encryption. The motivation of the work is clear that the previous approach such as using a session key obtained through the Diffie-Hellman key exchange algorithm can be improved in terms of network bandwidth and end-to-end delay.
Even though this paper is addressing a timely topic, which is one of the most important for vehicular communication, it is still questionable if this approach can be a more suitable solution than existing approaches.
- The authors claim that the existing scheme needs a long time to establish a session key. The proposed scheme also requires a vehicle to communicate with the trustworthy cloud for vehicle status information, but this has not been considered.
- It is not quite clear why this approach is the Quantum-Inspired Encryption.
- The control messages communicated with the trustworthy cloud also incurs the network bandwidth overhead and communication delay. This topic need to be included in the simulation results.
- How does proposed system differ from the old-fashioned centralized system?
- It is an interesting idea to utilize the trustworthy cloud to avoid key negotiation process between vehicles, however, the paper did not provide sufficient results to support the claim that the proposed scheme performs better than existing approach in vehicular networks. This should be addressed to improve the quality of the paper.
Author Response
Point 1: The authors claim that the existing scheme needs a long time to establish a session key. The proposed scheme also requires a vehicle to communicate with the trustworthy cloud for vehicle status information, but this has not been considered.
Response 1: In our scheme, vehicles communicate with the cloud to get the encryption key, and each vehicle has its own authentication certificate to ensure identity information. Using the difference of vehicle status information in time dimension in the cloud, we create message encryption keys, so that the security communication for VANETs does not need to establish a security alliance beforehand, nor need to negotiate keys, nor need to verify the packet replay during the interaction. In a communication process, the communicated vehicles interact with cloud by LTE channel only once respectively, saving a lot of repeated interaction steps.
In the subsection of negotiation-free encryption method, we mention that LTE technology is used between vehicle-cloud communications and LTE is very beneficial for delay-sensitive vehicular applications. LTE can provide a round-trip time theoretically lower than 10 ms, and transfer latency in the radio access up to 100 ms.
Point 2: It is not quite clear why this approach is the Quantum-Inspired Encryption.
Response 2: The title is changed to “Negotiation-free Encryption for Securing Vehicular Unicasting Communication”.
The reason why the title was called Quantum-inspired before is that our method has some similarities with the behavioural characteristics of quantum key distribution. They both generate and share a random and secure key to encrypt and decrypt messages. The most common correlation algorithm for quantum key distribution is one-time cryptography. We propose an encryption method, in which the key is tightly coupled with the vehicle status stored on the vehicle cloud. As time goes by, the state of the vehicle will change and the generated key will expire. However, quantum communication technology is not used in the process of our encryption system, so we change the title to avoid confusion.
Point 3: The control messages communicated with the trustworthy cloud also incurs the network bandwidth overhead and communication delay. This topic needs to be included in the simulation results.
Response 3: According to our scheme, the control message transmitted between the vehicle and the cloud is only used when both sides apply for the encryption key from the vehicle cloud, and the control message only needs to be transmitted once at both sides of the communication. After obtaining the key, the two sides communicate with each other directly to transmit the data package, instead of repeatedly communicating with the vehicle cloud. Thus, reducing the number of control message communications between vehicle and cloud can reduce the time delay and bandwidth overhead.
In the subsection of comparison and analysis of two methods, we use GNS3 to evaluate the NFEM and traditional IPSec encryption methods, and compare them with existing encryption methods in terms of key distribution delay and bandwidth overhead. The test results are shown in tables 4 and 5. The comparative data illustrate the delay and bandwidth overhead of several encryption methods, and the delay includes the control message transmission. The test data show that NFEM reduces the bandwidth and delay overhead, and has better performance than traditional encryption methods for VANET.
Point 4: How does proposed system differ from the old-fashioned centralized system?
Response 4: In NFEM, the existing vehicle status information stored on the cloud can be used to generate the encryption key for each vehicle communication session, and the vehicle cloud is regarded as a key distribution center, with no need to build a new key distribution server. Moreover, the performance of current cloud server is also far superior to that of traditional centralized key distribution servers.
From the process point of view, the traditional KDC generates the encryption key for the communication session, and encrypt and send it to the requester. Then the requester transfers the key to the respondent. When the respondent receives the encryption key, it also needs to send a reply to the requester to verify that the received communication request is true. During this period, there are several interaction steps between the two sides of the communication, and there exists a risk of being attacked by a third party. In the NFEM method, the cloud server generates the encryption key by hashing the vehicle status information of the respondent at a certain time, and returns the key to the both communicating parties separately. Based on the timestamp, the respondent obtains the encryption key directly from the vehicle cloud, decrypts the ciphertext, which avoids the extra interaction steps of KDC and the third party attack risk.
In the part of conclusions, we add these discussions to clarify the difference between NFEM and the old-fashioned KDC system.
Point 5: It is an interesting idea to utilize the trustworthy cloud to avoid key negotiation process between vehicles; however, the paper did not provide sufficient results to support the claim that the proposed scheme performs better than existing approach in vehicular networks. This should be addressed to improve the quality of the paper.
Response 5: Using the vehicle status stored in the cloud to generate encryption keys, a principle is need to be followed, that is the vehicle initiating secure communication could only request the current status information of other vehicles, but can not request the historical information of other vehicles. Thus the confidentiality of the encryption keys can be ensured.
We compare the characteristics of several encryption algorithms of NFEM and IPSec, and add Table 2 to show the superiority of NFEM. Furthermore, the performance of the NFEM method in terms of delay is statistically compared under different vehicle density, vehicle speed and outsourcing rate, as shown in Figures 6, 7, 8 and 9, which shows that the proposed method is superior to the existing encryption method, and is reliable and stable.

Reviewer 2 Report
The paper fails in one very important aspect. As it is a work for vehicular communications, the LTE sidelink should be referenced and the article has no reference on this topic. LENA is a very tool for SON, but I have serious doubts if is the best platform to test vehicular communications and cryptography.
The conclusion should be improved and reflect the achievements at work.
Author Response
Point 1: The paper fails in one very important aspect. As it is a work for vehicular communications, the LTE sidelink should be referenced and the article has no reference on this topic.
Response 1: In the related work section, besides the two original ones [23][24], we add two extra references about the application of LTE in vehicle communication,which are [25] and [26]. [25] describe the long-term evolution-vehicle (LTE-V) standard for sidelink or V2V communications Especially, [26] introduce the European AutoMat project defining an open Common Vehicle Information Model (CVIM), and describe a car-to-cloud communication model leveraging LTE uplink channels.
Point 2: LENA is a very tool for SON, but I have serious doubts if is the best platform to test vehicular communications and cryptography.
Response 2: In the security architecture of NFEM for vehicle unicast communication proposed by us, LTE is mainly used to transfer request packets and response keys packets between mobile vehicles and clouds. In the experiment, we use LENA module under NS-3 to measure the delay of the process of key request and return, to prove the superiority of our proposed method. There exists some other simulation platforms such as TraNS, MoVES, etc. that can be used to simulate the VANET environment, but the open source LENA module is more applicable in our scenario. In the simulation environment subsection, we add some introduction of simulator choose and comparison.
Point 3: The conclusion should be improved and reflect the achievements at work.
We improve the conclusion as shown in the paper and the following.
This paper proposes the negotiation-free encryption method (NFEM) based on VANETs. In NFEM, the existing vehicle status information stored on the cloud can be used to generate the encryption key for each vehicle communication session, and the vehicle cloud is regarded as a key distribution center, with no need to build a new key distribution server. Moreover, the performance of cloud is also far superior to that of traditional centralized key distribution servers.
From the process point of view, the traditional KDC generates the encryption key for the communication session, and encrypt and send it to the requestor. Then the requestor transfers the key to the respondent. When the respondent receives the encryption key, it also needs to send a reply to the requestor to verify that the received communication request is true. During this period, there are several interaction steps between the two sides of communication, and there exists a risk of being attacked by a third party. In the NFEM method, the cloud server generates the encryption key by hashing the vehicle status information of the respondent at a certain time, and returns the key to the both communicating parties separately. Based on the timestamp, the respondent obtains the encryption key directly from the vehicle cloud, decrypts the ciphertext, which avoids the extra interaction steps of KDC and the third party attack risk.
Compared with the traditional encryption methods of self-organizing network, NFEM mainly has the following characteristics. (1) In the transmission encryption process, it can reduce the key agreement, delay of encryption and overhead of key management. (2) The state value of vehicle attribute is used as a key, rather than by calculation and consultation with some of the key independent attributes of the vehicle itself, which greatly reduces the key generation time and meets the application requirements of VANETs. (3) In contrast with the traditional consultative mechanisms, NFEM reduces bandwidth consumption. In complex distributed vehicles network environments, as the vehicle density increases, the minimized bandwidth usage will be more favourable to network transmission. (4) In terms of security, the invariability of vehicle cloud and timestamp can ensure the security and privacy of data communication and control transmission for VANETs. Hence, our next work will be focused on transportation security.

Round 2
Reviewer 1 Report
The authors addressed the suggestions mentioned in the previous comments.